# Biocontrol Potential of the New Codling Moth Granulovirus (CpGV) Strains

**DOI:** 10.3390/microorganisms12101991

**Published:** 2024-09-30

**Authors:** Aleksandra A. Tsygichko, Anzhela M. Asaturova, Tatiana N. Lakhova, Alexandra I. Klimenko, Sergey A. Lashin, Gennady V. Vasiliev

**Affiliations:** 1Federal State Budgetary Scientific Institution, Federal Research Center of Biological Plant Protection, Krasnodar 350039, Russia; biocontrol-vniibzr@yandex.ru; 2Institute of Cytology and Genetics, Siberian Branch of Russian Academy of Sciences, Novosibirsk 630090, Russia; tlakhova@bionet.nsc.ru (T.N.L.); klimenko@bionet.nsc.ru (A.I.K.); lashin@bionet.nsc.ru (S.A.L.); genn@bionet.nsc.ru (G.V.V.)

**Keywords:** *Cydia pomonella*, CpGV, baculoviridae, granulovirus identification, entomopathogenic activity, virus insecticide

## Abstract

The use of CpGV strains as the basis for bioinsecticides is an effective and safe way to control *Cydia pomonella*. The research is aimed at the identification and study of new CpGV strains. Objects of identification and bioinformatic analysis: 18 CpGV strains. Sequencing was carried out on a NextSeq550. Genome assembly and annotation were carried out using Spades, Samtools 1.9, MinYS, Pilon, Gfinisher, Quast, and Prokka. Comparative genomic analysis was carried out in relation to the reference genome present in the «Madex Tween» strain-producer (biological standard) according to the average nucleotide identity (ANI) criterion. The presence/absence of IAP, cathepsin, MMP, and chitinase in the genetic sequences of the strains was determined using simply phylogeny. Entomopathogenic activity was assessed against *C. pomonella* according to the criterion of biological efficacy. Thus, molecular genetic identification revealed that 18 CpGV strains belong to a genus of Betabaculovirus. For all the strains under study ANI values of 99% or more were obtained, and the presence of the cathepsin, chitinase, IAP, and MMP genes was noted. The strains BZR GV 1, BZR GV 3, BZR GV 7, BZR GV 10, and BZR GV L-8 showed the maximum biological efficacy: 100% on the 15th day of observation. Strains BZR GV 4, BZR GV 8, and BZR GV 13 showed efficacy at the level of the «Madex Tween» preparation: 89.5% on the 15th day of observation. The strains with the highest mortality rate of the host insect were identified: BZR GV 9, BZR GV 10, BZR GV L-6, and BZR GV L-8.

## 1. Introduction

One of the methods for controlling the number of insect pests on agricultural crops is the use of insecticides based on entomopathogenic baculoviruses. Strains of nuclear polyhedrosis virus (NPV) or granulovirus (GV) are often used as the basis for preparations [1,2]. Such preparations are highly effective against a target insect, more narrowly focused compared to chemical insecticides, and safe against other species in the ecosystem [3]. The efficacy of viral insecticides against a target insect can reach 83–99% [4,5].

*Cydia pomonella* granulovirus (CpGV) or *Betabaculovirus cypomonellae* is an agent that has a worldwide distribution present in bioinsecticides used against the codling moth *Cydia pomonella* L., 1758. Thus, based on various strains of CpGV, there are products such as «Madex Top», «Carpovirus Plus», «Carpovirusine», «CYD-X», and many others [1,6,7]. According to Sauer et al. [8], while baculovirus insecticides are used, natural populations of insects can develop resistance, which is also confirmed by Haase et al. [9]. It can be assumed that one of the main factors in the resistance development is excessively long-term application of the same preparation without changing the active base strain. According to Eberle et al., as well as a published review by Siegwart et al., it is indicated that with constant exposure of the pest population to the preparation, most individuals die, but the remaining ones can acquire immunity to the insecticide [10,11]. That is why it is necessary to constantly search for new strains of entomopathogenic viruses with insecticidal properties. Studying the properties of baculovirus CpGV isolates will not only expand the existing databases of beneficial microorganisms but also help develop highly effective bioinsecticides for plant protection based on them in the future.

The search, isolation, and study of the properties of new CpGV strains capable of controlling the number of *C. pomonella* is certainly important for the entire world community. Especially for countries producing food, including apples. The main suppliers of apples in Europe are Poland, Spain, and France, with an area of apple plantations of about 470 thousand hectares [12,13]. In Russia, this figure is about 220 thousand hectares [14]. In addition to apples, *C. pomonella* can also damage pear fruits and walnuts [15]. It is worth noting that under favorable conditions for *C. pomonella*, it is capable of having 3–4 generations per season, increasing the level of damage up to 70.0% or more [16,17]. The use of baculoviruses to control the number of phytophages will help not only preserve the crop but also obtain environmentally friendly products without the use of chemical insecticides [18]. It is worth noting that in 2024 there are only two baculovirus preparations on the plant protection market in Russia: «Karpovirusin, SK» (CpGV isolate M, 1.0 × 10^13^ viral occlusion bodies/L, «Arysta LifeScience Corporation», Belgium) and «Helicovex, SK» (HearNPV, 7.5 × 10^12^ viral occlusion bodies/L, «Andermatt Biocontrol Suisse AG», Switzerland), therefore there is a need to expand the range of insecticides based on entomopathogenic baculoviruses [19].

Thus, the research is aimed at the identification and exploration of new CpGV strains according to the criteria of entomopathogenic activity against *C. pomonella*. For the first time, we characterized entomopathogenic virus strains indigenous to the south of Russia, which have potential as a bioinsecticide strain-producer against insects of the natural *C. pomonella* population.

## 2. Materials and Methods

### 2.1. Viruses and Insects

The objects of this study were 18 CpGV strains from the bioresource collection of the Federal State Budgetary Scientific Institution «Federal Research Center of Biological Plant Protection» «State Collection of Entomocarifages and Microorganisms» (BRC FSBSI FRCBPP). BRC FSBSI FRCBPP was registered on the basis of the Federal State Budgetary Scientific Institution «Federal Research Center of Biological Plant Protection» (FSBSI FRCBPP) in 2018 under the supervision of A.M. Astaturova. The collection contains a number of entomophages, phytopathogens, and microorganisms with various properties.

In this research, we used the scientific equipment «Technological line for obtaining microbiological plant protection products of a new generation». The strains were isolated from different stations in Kazakhstan and Russia in the Appendix A. All CpGV strains were stored at +4 °C in a refrigerator until use [20,21]. The target insect was a natural *C. pomonella* population. Larvae of the natural population of *C. pomonella* were collected in the amount sufficient to carry out the experiment in fruit plantations (domestic apple tree—*Malus domestica* Borkh., 1803) on the territory of the FSBSI FRCBPP. After collection, insects were kept in quarantine for 3–5 days using Mitrofanov artificial nutrient medium (Table 1) [22].

### 2.2. Preparation Procedure

We placed all dry ingredients (malt sprouts, corn flour, wheat bran) in a drying cabinet for 30–40 min at a temperature of 45–50 °C. The agar was diluted with 300 mL of water and dissolved in a water bath (5–7 min). Then we poured 300 mL of water into the soy flour and cooked in a water bath (with stirring) for 20 min. When mixing, we placed all the heated dry ingredients into the homogenizer and added the hot soy flour mixture. With constant stirring, we added melted agar-agar and mixed again. Then citric and sorbic acids were added and mixed. The temperature of the artificial medium did not fall below 50 °C. Lastly, we added ascorbic acid and methylparaben. At the end, the remaining hot water was added. Afterwards, we poured the medium into the required containers. The diet was ready for consumption. It was stored at a temperature of 4 °C for about 1–2 months.

### 2.3. Molecular Genetic Identification of CpGV Strains

During the molecular genetic identification of the CpGV strains under study, total DNA was isolated using the PureLink™ Genomic DNA Mini Kit Invitrogen (Missouri, TX, USA). During the isolation process, we did not use the methods of dissolving granulin, since during the research we discovered a fraction of viral DNA of sufficient length necessary to work with it. Ultrasonic DNA fragmentation was carried out on a Covaris M220 device with parameters optimized to obtain a maximum of 300 bp fragments. The resulting fragments were purified by sorption on magnetic particles with size selection to remove the fraction less than 150 bp. Amplification of libraries was carried out during 9 PCR cycles, followed by deposition on magnetic particles. The quality of the libraries was determined using a BA2100 bioanalyzer. Sequencing of the resulting libraries was carried out on a NextSeq550 device with paired-end reads of 2 × 150 bp. according to the manufacturer’s protocol. The quality of sequencing results was checked using the FastQC program (v. 0.11.9) [23]. Genome assembly was carried out using Spades (v. 3.13.0) [24,25], Samtools (v. 1.9) [26,27], MinYS (v. 1.1) [28], Pilon (v. 1.24) [29], and Gfinisher (v. 1.4) [30] programs. The quality of the assemblies was assessed using the Quast (v. 5.2.0) program [31]. Genome annotation was performed using the Prokka (v. 1.14.6) tool [32] with standard parameters, except for the—Kingdom parameter responsible for annotation selection, which was specified as “viruses”. The resulting assemblies and annotations of 18 CpGV strains have been deposited in the NCBI database according to standard protocols. The accession numbers are shown located in Appendix A.

### 2.4. Comparative Genomic Analysis

Comparative genomic analysis of 18 CpGV strains under study was carried out against the reference genome isolate Mexico/1963 or CpGV-M or NC_002816.1 (https://www.ncbi.nlm.nih.gov/nuccore/NC_002816 accessed on 10 April 2024) from the NCBI database and the genome of the standard preparation «Madex Tween» strain-producer [33], which was obtained by repeated isolation of viral DNA from a commercial product according to the method indicated above. Average Nucleotide Identity (ANI) between genomes was calculated using the FastANI program (v. 1.1) [34] with the following parameters: –ql, –rl, –minFrag 35 –output, where –ql was responsible for the list of supplied genomes that need to be compared, –rl was responsible for the list of genomes with which we want to compare, --minFrag meant the number of fragments into which the genome will be split. By default, it is 50. There is also a fragLen parameter, which is responsible for the length of the fragment (3000 by default). For granuloviruses, these two parameters were adjusted because the total length of the genome sequences is not enough for the default parameters. In this case, the length of the fragment was saved, but their number was reduced. The –output parameter wrote a text file with genome similarity values. Visualization of the nucleotide identity of the strains was carried out using a heat map of genome similarities using the gplots library of the R programming language.

### 2.5. Phylogenetic Analysis

A phylogenetic tree was constructed based on 18 obtained fragmented assemblies of CpGV genomes as well as Madex Twin. In this case, CpGV-M (KM217575.1) and NC_002816.1 were taken as reference genomes. *Thaumatotibia leucotreta* granulovirus (CrleGV) (NC_005068) was used as an outgroup.

Multiple alignments were performed using the MAFFT program (v. 7.525) [35]. Because the reference assembly starts at orf-1 encoding granulin, CpGV assemblies were also reorganized relative to this start for more correct alignment.

The assemblies were also compared to the reference assembly using BWA (v. 0.7.17) [36]. To identify contigs carrying redundant information. Such fragments were removed from the assemblages before reorganization for alignment.

A phylogenetic tree was constructed using the maximum likelihood method and the Jukes–Cantor model. A bootstrap support of 500 cycles was used. The tree was constructed in the MEGA 11 program [37].

### 2.6. Entomopathogenic Activity

The entomopathogenic activity of 18 CpGV strains was assessed against L2–L5 larvae of *C. pomonella* in vitro. It is worth noting that in this work we studied a natural population of insects collected from the environment and not a laboratory population. This fact complicates the selection of the required number of individuals of one age group, so we use insects of different ages in the study. Aqueous viral suspensions, which were prepared immediately before the experiment, were used as an inoculant. The stages of preparing aqueous viral suspensions included the myceration of the infected biomaterial, filtration, and centrifugation of each sample at 3000 rpm for 10 min, followed by resuspension [20,21]. Quantitative assessment of the virus was carried out using a microscopic imaging system, Carl Zeiss, Axio Scope A1, (Germany) at a magnification of ×400 using a Goryaev counting chamber, which is an analogue of the Neubauer chamber (Neubauer hemocytometer, China). The preparation «Madex Tween» (CpGV isolat V22, 3.0 × 10^13^ viral occlusion bodies/L, «Andermatt Biocontrol Suisse AG», Switzerland) was used as a biological standard [38]. At the time of the research, this preparation was registered and approved for use in Russia against *C. pomonella*. It must be emphasized that the working solutions of the virus strains had different concentrations. This can be explained by the fact that the main goal of the study was not to search for an effective concentration but to confirm/determine the entomopathogenic activity of the strains against the object. We plan to carry out studies of the effective concentration of strains at the next stages of the study. Mitrofanov’s artificial nutrient medium was used as a diet [22]. Inoculation was carried out by surface contamination of the diet [39]. In the experimental options, 0.03 mL of an aqueous viral suspension of the test strain was treated per 1 g of diet; in the standard option, −0.03 mL of the working solution of the preparation; in the control option, −0.03 mL of distilled water. The concentrations of working solutions are presented in Table 2. During and after feeding, insects were kept individually in wells in plastic sectional plates «Falcon, SPL Lifesciences» in a climate chamber Binder KBWF 720 (Germany) at a temperature of +26 °C, air humidity 70%, and a photoperiod of 18:6 h (day/night) [39,40,41]. In each analysis, 5 larvae were used, the experiment was repeated nine times. Insect mortality was determined on days 1, 2, 3, 5, 7, 10, and 15 after inoculation.

### 2.7. Data Analysis

Statistical analysis was carried out using the Statistica 12 program. Mathematical processing of the obtained data was carried out using standard computer programs (Microsoft Excel). Normal distribution was checked using the Kolmogorov–Smirnov test (*p* < 0.05). Comparison between options was carried out using Duncan’s test at a 95% probability level, *p* = 0.05. Entomopathogenic assessment of the CpGV strains efficacy under laboratory conditions was carried out using the Henderson–Tilton biological efficacy formula [42]. The resulting negative values were equated to zero efficiency. Insect survival was analyzed using the Kaplan–Meier method with a confidence interval of 95% or more, and the average and median time of death were determined [43].

## 3. Results

### 3.1. Molecular Genetic Identification of CpGV Strains

In the course of the research, the DNA of 18 CpGV strains was isolated from the BRC of the FSBSI FRCBPP in a volume of 50 μL per sample. The 150–400 bp fraction was found to account for 4.5–10% of total DNA. The volume of the obtained libraries of the studied strains ranged from 22.1 million to 36.2 million paired reads, which made it possible to assemble genomes with a coverage of more than 500×. When assessing the quality of the assemblies, it was revealed that the majority of genome samples contained two peaks of GC composition. Thus, the assemblies of the studied genomes were similar in length to the reference genome. All 18 strains underwent the deposition procedure in the international NCBI database and were assigned individual numbers, which are given in the Appendix A. Since the genome assemblies were able to assemble up to several sequences, each genome was assigned from one to 28 numbers.

### 3.2. Genomic Analysis Based on Average Nucleotide Identity

During the comparative genomic ANI analysis of 18 CpGV strains against the «Madex Tween» strain-producer (biological standard) and two references: NC_002816.1 and KM217575.1 (references from NCBI), color scales were assigned, the interval values of which were selected based on the obtained ANI values (Figure 1).

ANI values of 99% or more were obtained for all 18 CpGV strains against the genome of the «Madex Tween» strain-producer and the reference genomes NC_002816.1 and KM217575.1. The highest estimate of average nucleotide identity for NC_002816.1 and KM217575.1 have been found in the BZR GV 4 and BZR GV 6 strains, which amounted up to ANI = 99.8477% and ANI = 99.8682%. In addition, another strain, BZR GV L-7, has high ANI values with respect to KM217575.1. Its value is 99.8862%. The highest estimate of average nucleotide identity for «Madex Tween» was found in the BZR GV 4, BZR GV 6, and BZR GV 7 strains, which amounted to ANI = 99.9041%, ANI = 99.9876%, and ANI = 99.8920%, respectively. 

In this case, for all three genomes, both reference and from the commercial product, the four genomes: BZR GV L-4, BZR GV L-5, BZR GV L-6, and BZR GV L-8, have the lowest ANI scores.

### 3.3. Phylogenetic Tree

We decided to see how close the investigated CpGV genomes are to each other and in relation to the reference genome, as well as to the genome derived from the commercial Madex Twin preparation.

A phylogenetic tree has been constructed based on the obtained fragment assemblies shown in Figure 2. The CrleGV genome (NC_005068) has been used as an outgroup. The phylogenetic tree has been constructed and visualized in the MeGA 11 program [37].

Here we see the correspondence obtained by calculating the ANI values. The strain BZR GV 6 showed the highest ANI value in relation to the references and Madex Twin, and here we see that it is classified in the same clade with them. Four genomes: BZR GV L-4, BZR GV L-5, BZR GV L-6, and BZR GV L-8, were identified as a separate clade.

### 3.4. Entomopathogenic Activity

A comparative assessment of the entomopathogenic activity of 18 CpGV strains with the standard «Madex Tween» was carried out using viral suspensions, the titer of which ranged from 8.0 × 10^6^ to 1.3 × 10^8^ granules/mL. The titer of the working solution of the biological standard is 3.0 × 10^6^ granules/mL (Table 2). It should be noted that in Table 2, there are no statistically significant differences between the options designated by the same letters when compared within the columns according to the Duncan test at the 95% probability level, *p* = 0.05.

It was found that the biological efficacy of the studied strains varied from 0 to 64.2%, and of the biological standard—53.3% on the fifth day, which indicates its high insecticidal activity on the first day after application. It is worth highlighting the BZR GV L-8 strain, with the activity of 64.2%, which is 10.9% higher than that of the standard one. It was revealed that on the 15th day the biological efficacy of the studied strains varied from 65.2 to 100%, and of the biological standard, 89.5%, which corresponds to the declared level. It is necessary to highlight the BZR GV 1, BZR GV 3, BZR GV 7, BZR GV 10, and BZR GV L-8 strains, the biological efficacy of which was 100%, which is significantly higher than in the “Madex Tween” option, as well as the BZR GV 6 strain with the insecticidal activity of 93.0%. The efficacy in the BZR GV 4, BZR GV 8, and BZR GV 13 strain options was at the standard level: 89.5%.

It was noted that with the same level of biological efficacy in options with individual strains, the death of insects was observed throughout the entire period of the study, and in some options only at the end of the experiment. For example, the death of insects in the BZR GV 5 strain option was observed over 15 days, but with the BZR GV 13 strain, the manifestation of insecticidal activity was observed only from the seventh day of the experiment.

In the process of analyzing the survival of *C. pomonella* larvae, it was found that, according to the criterion of average death time, the most effective strains were BZR GV 1, BZR GV 7, BZR GV 9, BZR GV 10, BZR GV L-6, and BZR GV L-8, with the time ranged from 4.6 to 6.6 days. According to the criterion of median death time, the most active strains were BZR GV 9, BZR GV 10, BZR GV L-6, and BZR GV L-8, their range was from three to five days. With a comprehensive assessment of the average and median time of death, the strains with the highest mortality rate were identified: BZR GV 9, BZR GV 10, BZR GV L-6, and BZR GV L-8; their activity was at the standard level or slightly lower (Table 3).

## 4. Discussion

This research has identified and studied the entomopathogenic properties of 18 CpGV strains from the BRC of the FSBSI FRCBPP, isolated from various stations in the south of Russia and Kazakhstan. In accordance with modern standards, molecular genetic assessment methods were used to identify entomopathogenic baculoviruses, as well as ANI analysis in relation to the reference genome and the genome of the strain isolated from the commercial product «Madex Tween». To study the entomopathogenic properties of granulovirus strains, a comprehensive assessment of biological efficacy and lethality time was carried out, which made it possible to identify the most active samples.

In the process of molecular genetic identification at the stage of preparation of genomic libraries, DNA fractions of more than 150 bp were isolated, which made it possible to eliminate impurities of highly degraded host DNA and degraded DNA of the CpGV strains under study. It should be noted that usually when isolating genetic material, the amount of degraded DNA is about 4.5–10%. However, the detected high level of virus degradation (90–95%) can be explained by the fact that during long-term storage of biomaterial in the form of an aqueous viral suspension, active decomposition of virus granules occurs and/or noticeable contamination of the material with degraded host DNA adhering to the surface of viral particles. Therefore, it is necessary to pay attention to finding the optimal storage method for CpGV collection strains in future studies. Thus, in their work, to increase the shelf life of baculovirus cultures, researchers use glycerin, liquid nitrogen, various artificial media, freezing, lyophilization, etc. [44,45]. It should be noted that in this study we did not assemble strains to the ring chromosome level but only partial sequences.

During the quality assessment of 18 CpGV assemblies and the assembly of the bioinsecticide «Madex Tween» strain-producer, the presence of two peaks of GC composition was revealed. This effect was caused by the presence of host insect DNA in the samples, which confirmed earlier results about the presence of degraded genetic material. Thus, the real GC composition of readings began to correspond to the theoretical one. Thus, we were convinced of our assumption about the presence of non-viral DNA in the raw data, which is confirmed by Fraser et al. [46].

In the course of the comparative genomic analysis, the ANI values >99% were obtained for all 18 genomes of CpGV strains in relation to the genome of the bioinsecticide «Madex Tween» strain-producer and the reference genome NC_002816.1. The assemblies of the studied genomes are similar in length and composition to the reference, which indicates that they belong to the Betabaculovirus cypomonellae. The greatest similarity to the reference and genome of the bioinsecticide strain-producer was found in the BZR GV 6 strain at 99.9%. It can be assumed that the entomopathogenic properties and mechanisms of action against insects of the BZR GV 6 strain will be similar to those of the reference and strain-producer, but this issue requires more detailed study in the future.

When analyzing the phylogenetic tree of 18 CpGV strains in comparison with Madex Twin and reference strains, we fully confirmed the results of the ANI analysis obtained previously. It can be assumed that the genetic proximity of the strains is due not only to their belonging to Betabaculovirus cypomonellae but also to the presence in the genetic apparatus of “insecticidal” proteins that are involved in the process of insect virosis. Thus, the presence of cathepsin, chitinase, IAP, or MMP was found in all 18 CpGV strains. Cathepsin is an enzyme involved in the homogenization of internal insect tissues; chitinase is an enzyme that causes the destruction of the chitin cover; IAP is an apoptosis inhibitor involved in the implementation of programmed cell death; and MMP is a matrix metalloprotease of the family of zinc-dependent endopeptidases that destroy extracellular matrix proteins [47,48,49]. The data obtained during the research indicate the presence of all four proteins in each of the studied strains. It can be assumed that the biological efficacy of CpGV strains directly depends on the presence of “insecticidal” proteins in their composition, their location, and various combinations with each other. The presence of these proteins in all CpGV strains under study makes them a valuable asset as a basis strain for the development of highly effective bioinsecticides for the protection of fruit cenoses against *C. pomonella*.

When assessing the biological efficacy of 18 CpGV agents against the natural *C. pomonella* population, it was found that the BZR GV 1, BZR GV 3, BZR GV 7, BZR GV 10, and BZR GV L-8 strains showed maximum activity: 100% on the 15th day of observation. The BZR GV 4, BZR GV 8, and BZR GV 13 strains showed efficacy at the standard level: 89.5% on the 15th day of observation. It is worth noting that the efficacy of the biological standard is ensured not only by the base strain but also by the presence of additional components: preservatives, fillers, antioxidants, adsorbing agents, sunscreen components, food attractants, etc. [50,51,52]. Therefore, the selection of strains with activity at the level of a commercial product or more makes it possible to develop in the future a bioinsecticide based on them that is more effective than the standard one. Analysis of the obtained data using the Duncan criterion made it possible to determine the BZR GV 1, BZR GV L-6, BZR GV L-8, and «Madex Tween» options to be significantly different from the control from the first and/or second day of the experiment. The BZR GV 8, BZR GV 9, and BZR GV 10 strain options were significantly different from the control from the third day of the experiment. In the BZR GV 14 strain option, no significant differences from the control were detected. Thus, we can identify strains that are most promising for the development of a bioinsecticide against *C. pomonella* based on them: BZR GV 1, BZR GV 8, BZR GV 10, BZR GV L-8. It must be emphasized that in this study we did not find a relation between biological efficacy and the geographical origin of the strains. The results obtained indicate the high species specificity of the studied CpGV strains from the BRC of the FSBSI FRCBPP, so we can conclude that there are pronounced inter-strain differences, which are the individual characteristics of each sample.

When analyzing the survival rate of larvae of the natural *C. pomonella* population, the Kaplan–Meier method was used, which allows us to determine the proportion of surviving/dead individuals using the mortality rate expressed in terms of the average and median time of death. The average death time shows the average time period during which all individuals in the sample die. The median time of death makes it possible to determine the time required for the death of a certain % of insects (in this case, 50%) in the sample [53]. Therefore, the lower the average and median time of death of individuals in the population, the higher the mortality rate of insects and the more effective the insecticidal preparation/strain is. Thus, it becomes possible to assess strains not only by the criterion of biological efficacy but also by the criterion of their speed of action. Thus, during the study, it was noted that the standard had the highest mortality rate on the first day of the experiment. It quickly destroyed half the population, which was expressed in a median time of death of 2.0 days. In turn, the BZR GV L-8 strain was the best option for the proportion between the initial rate of death and long-term efficacy since its median time of death was equal to 3.0 days, which indicates a slower effect than when using the biological product «Madex Tween»; however, the average time of death was at the level of 4.6 days, which is 0.9 lower than that of the biological standard. With a comprehensive assessment of the 18 studied CpGV strains based on the time of death of insects, we can conclude that they have a prolonged entomopathogenic effect against *C. pomonella*. Thus, in 66% of strains, the medial time of death is 7.0 days. In addition, there was no significant difference in the strains from Russia and Kazakhstan according to the above criterion.

It must be emphasized that, according to the published data, the use of entomopathogenic baculoviruses is possible not only against the target insect but also against closely related species [54,55]. It is also possible to combine viral insecticidal preparations with bacterial or fungal pesticides, which will increase their overall efficacy [3,56].

Thus, the potential of new, previously unknown CpGV strains may be much higher than that of existing cultures, which once again emphasizes the importance of working with them.

## 5. Conclusions

Thus, in the course of the research, 18 new CpGV strains with insecticidal activity against *C. pomonella* have been studied. Their molecular genetic identification and deposition made it possible to expand the list of available CpGV nucleotide sequences in the NCBI database. The data obtained are a promising basis for the development of highly effective, narrowly focused, and safe baculoviral insecticides in the future, which will contribute to the expansion of the market for plant protection products in Russia and other countries.

## Figures and Tables

**Figure 1 microorganisms-12-01991-f001:**
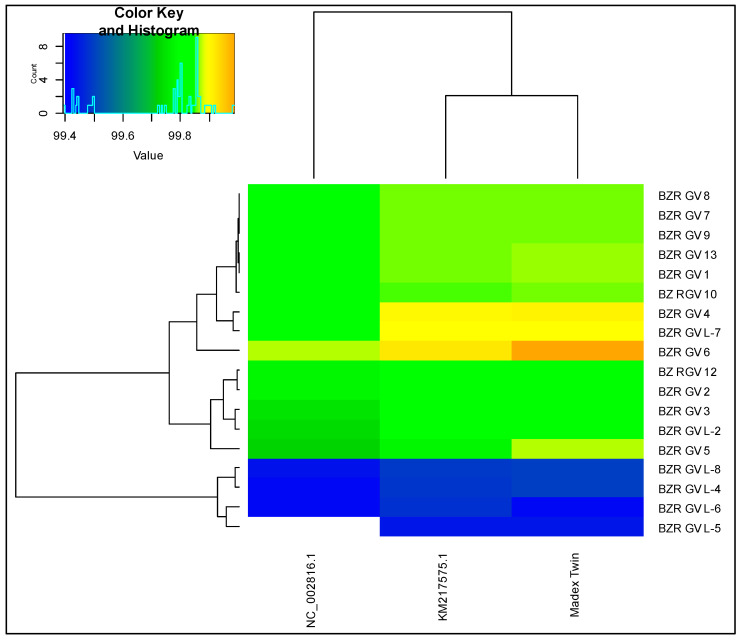
Heat map of similarities between genomes based on the fastANI result; below are the identifiers of references genomes CpGV: NC_002816.1, KM217575.1 and the «Madex Tween» strain-producer (biological standard); the identifiers of the 18 CpGV genomes under study are marked on the right side. Color scales of ANI values: from light blue to blue: (99–99.4%], from blue to green: (99.4–99.8%), from green to yellow: (99.8–99.90%), and from yellow to orange: (99.90–100%].

**Figure 2 microorganisms-12-01991-f002:**
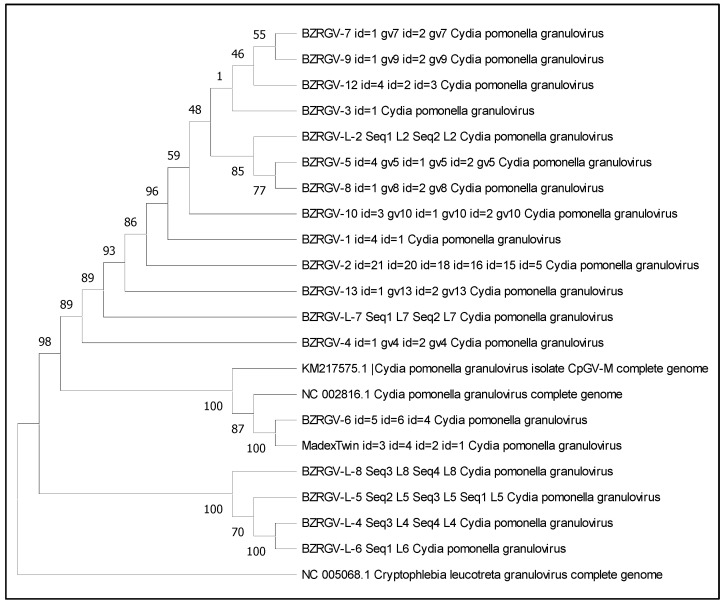
Phylogenetic tree of the 18 CpGV strains obtained relative to the reference CpGV-M and Madex Twin. CrleGV was selected as an outgroup. The tree was constructed using the maximum likelihood method and 500 bootstrap replicates. The Jukes–Cantor substitution model was selected. Here, the labels indicate the contig identifiers for the resulting CpGV genomes, which were used in multiple alignment and tree construction, respectively.

**Table 1 microorganisms-12-01991-t001:** Composition of Mitrofanov artificial nutrient medium for *C. pomonella*.

Ingredient	g or mL
Agar-agar	12
Corn flour	45
Wheat bran	45
Malt sprouts	54
Soy flour	40
Ascorbic acid	5.4
Citric acid	4.5
Sorbic acid	0.81
Methylparaben	2.25
Water	800

**Table 2 microorganisms-12-01991-t002:** Entomopathogenic activity of CpGV and «Madex Tween» strains against the natural *C. pomonella* population.

No.	Strain/Code	Titer, Granules/mL	Biological Efficacy, %, Days							Live Larvae, Ind., Days						
			1	2	3	5	7	10	15	1	2	3	5	7	10	15
1	BZR GV 1	(8.00 ± 3.00) × 10^6^	25.15	32.81	31.94	36.81	73.05	100.00	100.00	3.5 ^a,b^	3.0 ^b^	2.8 ^a,b,c,d^	2.5 ^b,c,d,e^	1.0 ^a,b^	0.0 ^a^	0.0 ^a^
2	BZR GV 2	(9.00 ± 2.00) × 10^6^	6.43	7.93	5.76	6.59	46.11	77.42	79.17	4.4 ^c,d,e^	4.1 ^d,e,f,g^	4.0 ^e,f,g,h^	3.7 ^f,g,h,i^	2.0 ^b,c,d,e,f^	0.7 ^a,b,c,d^	0.6 ^a,b,c,d^
3	BZR GV 3	(1.00 ± 0.40) × 10^7^	0	12.91	10.99	17.58	55.09	87.10	100.00	5.0 ^e^	3.8 ^b,c,d,e,f^	3.7 ^d,e,f,g,h^	3.3 ^c,d,e,f,g,h^	1.6 ^a,b,c,d,e^	0.4 ^a,b,c^	0.0 ^a,b^
4	BZR GV 4	(2.13 ± 0.32) × 10^7^	0	7.93	5.76	9.34	46.11	83.87	89.58	5.0 ^e^	4.1 ^d,e,f,g^	4.0 ^f,g,h^	3.6 ^e,f,g,h,i^	2.0 ^b,c,d,e,f^	0.5 ^a,b,c^	0.3 ^a,b,c,d^
5	BZR GV 5	(3.40 ± 0.79) × 10^7^	11.11	20.37	18.85	31.32	55.09	64.52	86.11	4.2 ^c,d^	3.5 ^b,c,d,e^	3.4 ^c,d,e,f,g^	2.7 ^b,c,d,e,f^	1.6 ^a,b,c,d,e^	1.2 ^b,c,d^	0.4 ^a,b,c,d^
6	BZR GV 6	(2.97 ± 2.02) × 10^7^	1.75	7.93	5.76	14.84	52.10	70.97	93.06	4.6 ^d,e^	4.1 ^d,e,f,g^	4.0 ^e,f,g,h^	3.4 ^d,e,f,g,h,i^	1.7 ^b,c,d,e,f^	1.0 ^a,b,c,d^	0.2 ^a,b^
7	BZR GV 7	(1.50 ± 0.20) × 10^7^	4.09	5.44	8.38	20.33	76.05	83.87	100.00	4.5 ^c,d,e^	4.2 ^d,e,f,g^	3.8 ^e,f,g,h^	3.2 ^c,d,e,f,g,h^	0.8 ^a,b^	0.5 ^a,b,c^	0.0 ^a,b^
8	BZR GV 8	(1.47 ± 0.41) × 10^7^	4.09	7.93	18.85	31.32	61.08	70.97	89.58	4.5 ^c,d,e^	4.1 ^d,e,f,g^	3.4 ^b,c,d,e,f^	2.7 ^b,c,d,e,f^	1.4 ^a,b,c^	1.0 ^a,b,c,d^	0.3 ^a,b,c,d^
9	BZR GV 9	(1.03 ± 0.25) × 10^7^	0	17.88	26.70	45.05	73.05	77.42	86.11	5.0 ^e^	3.6 ^b,c,d,e,f^	3.1 ^b,c,d,e,f^	2.2 ^a,b,c^	1.0 ^a,b^	0.7 ^a,b,c,d^	0.4 ^a,b,c,d^
10	BZR GV 10	(1.93 ± 0.05) × 10^7^	8.77	12.91	29.32	39.56	61.08	74.19	100.00	4.3 ^c,d^	3.8 ^c,d,e,f^	3.0 ^a,b,c,d,e^	2.4 ^a,b,c,d^	1.4 ^a,b,c^	0.8 ^a,b,c,d^	0.0 ^a,b^
11	BZR GV 12	(2.40 ± 0.52) × 10^7^	0	0	0	0	13.17	51.61	72.22	5.0 ^e^	5.0 ^g^	4.6 ^h,i^	4.5 ^i^	3.2 ^g,h^	1.6 ^d^	0.8 ^b,c,d^
12	BZR GV 13	(1.37 ± 0.32) × 10^7^	0	0	0	0	52.10	70.97	89.58	5.0 ^e^	5.0 ^g^	5.0 ^i^	4.5 ^i^	1.7 ^b,c,d,e,f^	1.0 ^a,b,c,d^	0.3 ^a,b,c,d^
13	BZR GV L-2	(2.70 ± 0.36) × 10^7^	0	0	0.52	12.09	67.07	74.19	79.17	5.0 ^e^	4.5 ^f,g^	4.2 ^g,h,i^	3.5 ^d,e,f,g,h,i^	1.2 ^a,b,c^	0.8 ^a,b,c,d^	0.6 ^a,b,c,d^
14	BZR GV L-4	(4.53 ± 2.91) × 10^7^	0	10.42	18.85	28.57	49.10	70.97	75.69	5.0 ^e^	4.0 ^c,d,e,f^	3.4 ^c,d,e,f,g^	2.8 ^b,c,d,e,f,g^	1.8 ^b,c,d,e,f^	1.0 ^a,b,c,d^	0.7 ^a,b,c,d^
15	BZR GV L-5	(6.77 ± 4.60) × 10^7^	1.75	10.42	5.76	28.57	58.08	77.42	82.64	4.6 ^d,e^	4.0 ^c,d,e,f^	4.0 ^f,g,h^	2.8 ^b,c,d,e,f,g^	1.5 ^a,b,c,d^	0.7 ^a,b,c,d^	0.5 ^a,b,c,d^
16	BZR GV L-6	(1.39 ± 0.19) × 10^8^	6.43	22.86	37.17	50.55	70.06	77.42	82.64	4.4 ^c,d,e^	3.4 ^b,c,d^	2.6 ^a,b,c^	2.0 ^a,b^	1.1 ^a,b,c^	0.7 ^a,b,c,d^	0.5 ^a,b,c,d^
17	BZR GV L-7	(9.00 ± 3.61) × 10^6^	15.79	15.39	13.61	12.09	28.14	58.06	65.28	4.0 ^b,c^	3.7 ^b,c,d,e,f^	3.6 ^d,e,f,g,h^	3.5 ^d,e,f,g,h,i^	2.6 ^d,e,f,g,h^	1.4 ^c,d^	1.1 ^c^
18	BZR GV L-8	(9.57 ± 1.71) × 10^7^	11.11	30.32	42.41	64.29	85.03	90.32	100.00	4.2 ^c,d^	3.1 ^b,c^	2.4 ^a,b^	1.4 ^a^	0.5 ^a^	0.3 ^a,b^	0.0 ^a,b^
19	«Madex Tween»	(3.08 ± 0.23) × 10^6^	27.49	50.23	50.26	53.30	64.07	67.74	89.58	3.4 ^a^	2.2 ^a^	2.1 ^a^	1.8 ^a,b^	1.3 ^a,b,c^	1.1 ^b,c,d^	0.3 ^a,b,c,d^
20	control	-	-	-	-	-	-	-	-	4.7 ^d,e^	4.4 ^e,f,g^	4.2 ^g,h,i^	4.0 ^g,h,i^	3.7 ^h^	3.4 ^e^	3.2 ^e^

**Table 3 microorganisms-12-01991-t003:** Analysis of *C. pomonella* survival when inoculated with CpGV and «Madex Tween» strains.

No.	Strain/Code	*p*-Value (Error Level)	Average Time of Death, Days	Median Time of Death, Days
1	BZR GV 1	<0.001	5.2	7.0
2	BZR GV 2	<0.001	7.7	7.0
3	BZR GV 3	<0.001	7.0	7.0
4	BZR GV 4	<0.001	7.6	7.0
5	BZR GV 5	<0.001	7.2	7.0
6	BZR GV 6	<0.001	7.8	7.0
7	BZR GV 7	<0.001	6.6	7.0
8	BZR GV 8	<0.001	7.0	7.0
9	BZR GV 9	<0.001	6.1	5.0
10	BZR GV 10	<0.001	6.5	5.0
11	BZR GV 12	<0.001	10.0	10.0
12	BZR GV 13	<0.001	8.7	7.0
13	BZR GV L-2	<0.001	7.6	7.0
14	BZR GV L-4	<0.001	7.4	7.0
15	BZR GV L-5	<0.001	7.2	7.0
16	BZR GV L-6	<0.001	5.8	5.0
17	BZR GV L-7	<0.001	8.5	10.0
18	BZR GV L-8	<0.001	4.6	3.0
19	«Madex Tween»	<0.001	5.6	2.0

## Data Availability

The data presented in this study are available on request from the corresponding author.

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
