# Peer review of "Biocontrol Potential of the New Codling Moth Granulovirus (CpGV) Strains"

_microorganisms, 2024, doi:10.3390/microorganisms12101991_

Round 1
Reviewer 1 Report
Comments and Suggestions for Authors
The manuscript by Tsygichko et al. describes the identification and exploration of new CpGV strains according to the criteria of entomopathogenic activity against C. pomonella. For this, the authors characterized entomopathogenic virus strains indigenous to the south of Russia, which have potential as a bioinsecticide strain-producer against insects of the natural C. pomonella population.
This an interesting study, the author used modern standards, molecular genetic assessment methods were used to identify entomopathogenic baculoviruses and founded a high similarity between the 18 genomes of CpGV strains in relation to the genome of the bioinsecticide «Madex Tween» strain-producer and the reference genome. However, my main concern focused on the methodology used for entomopathogenic activity. This section is very confusing, because the authors mentioned that entomopathogenic activity of 18 CpGV strains was assessed against larvae C. pomonella in vitro. Cleary is an in vivo technique (involving the infection of groups of insects). In addition, for the bioassays, the author used L2-L5 larvae collected from the field. However, the susceptibility of the host to baculovirus can vary greatly depending on age, among other factors. Therefore, I consider that the biological assays should be carried out for each larval age to determine the biological activity of the strains under study. I addition, the working solutions of the 162 strains and the biological standard had different concentrations. Therefore, this does not allow comparisons among them.
The authors determined that the biological efficacy of strains on L2-L5 C. pomonella
Author Response
Comment 1: Thank you very much for considering our publication. We agree with your comment. However, regarding the chapter "Bioanalyses", we would like to comment on why in this work we chose different concentrations of the virus and insects of different ages.
The experiment described in the publication is indicated as "in vitro", and not "in vivo", since the inoculation of insects, their maintenance and accounting were carried out under the controlled laboratory conditions and environmental factors did not affect C. pomonella in any way.
The design of the experiment was compiled according to standard microbiological methods of working with baculoviruses and methods of technical entomology (Dyck V. A., 2010;, 2011; Bayramoglu, Z., Nalcacioglu, R., Demirbag, Z. Et al, 2018 and others; Wennmann J.T., Pietruska D., Jehle J.A., 2021; Luque T., Finch R., Crook N., O'Reilly D.R., Winstanley D., 2001 and others).
We fully agree with you that for a correct comparison of entomopathogenic efficiency, it is necessary to use insects of the same age, as well as use solutions of the same concentration. However, in this study, the main goal was not to find an effective concentration, but to identify the very possibility of entomopathogenic activity of strains against C. pomonella. We are going to carry out studies to find an effective concentration of strains at the next stages of the research.
As for the age of the caterpillars, it is worth noting that in this research we studied a wild population of insects collected in natural environment. This fact complicates the selection of the required number of individuals of the same age group, so in the study we use insects of different ages. In addition, a feature of the development cycle of C. pomonella is that it can have 3-4 generations per season, which often overlap. As a result, the population simultaneously contains insects in the egg stage, caterpillars of different ages, as well as sexually mature individuals, which makes the selection of caterpillars of the same age especially labor-intensive.
It should be noted that at the next stage of our research we are going to assess the entomopathogenic activity of the most promising strains with the representation of a single concentration in the design of the experiment and the use of a laboratory population of C. pomonella
Reviewer 2 Report
Comments and Suggestions for Authors
Cydia pomonella is an insect that causes great damage to apple fruits, as well as pear fruits and walnuts. Baculovirus insecticides are used to combat them. However, these insects can develop resistance. Therefore, it is necessary to search for new strains of CpGV. This is the purpose of the present manuscript. Methodical studies were performed on 18 CpGV strains obtained from different stations in Kazakhstan and Russia. These strains were identified by molecular methods. The entomopathogenic activity of 18 CpGV strains was assessed against L2-L5 larvae of C. pomonella in vitro. The results are presented clearly. An important result obtained during the studies is the demonstration of the most effective entomopathogenic strains. The discussion is interesting. This manuscript should be published in Microorganisms after correcting the errors given in Remarks.
Remarks
Line 79 FSBSI FRCBPP – this abbreviation requires explanation
Line 83 – it should be here - ‘properties.’
Line 88 Larvas – it should rather be Larvae
Line 92 no dot at the end of the sentence
Line 93 g or mg - should be clarified in Table 1
Line 93 Table 1 Preparation procedure should be described in the text under the table, not in Table 1. This text cannot be given in the imperative mood.
Line 111 – consider revising this sentence
Line 136 Cryptophlebia leucotreta - should be checked, the current name of this species is Thaumatotibia leucotreta Meyrick. The species name should be written in italic.
Line 148 it should be ‘of C. pomonella’ ?
Line 251 C. pomonella - is written twice, this should be corrected
Line 250 text in some places in Table 2 needs to be corrected
Line 442 Cydia pomonella - this should be written in italics (also in other places in the Literature
Comments on the Quality of English Language
see Remarks
Author Response
Comment 1: Thank you very much for considering our publication. We fully agree with all your comments and remarks. All the remarks have been corrected in the text of the manuscript.
We have also provided the responses to the comments below:
Remarks:
Line 79 FSBSI FRCBPP – this abbreviation requires explanation
- Response to the comment: corrected in the text, deciphered the abbreviation.
Line 83 – it should be here - ‘properties.’
- Response to the comment: changed in the text to the correct wording.
Line 88 Larvas – it should rather be Larvae
- Response to the comment: changed in the text to the correct wording.
Line 92 no dot at the end of the sentence
- Response to the comment: the dot is put.
Line 93 g or mg - should be clarified in Table 1
- Response to the comment: corrected in the text for «g or ml».
Line 93 Table 1 Preparation procedure should be described in the text under the table, not in Table 1. This text cannot be given in the imperative mood.
- Response to the comment: corrected the table.
Line 111 – consider revising this sentence
- Response to the comment: changed in the text to the correct wording.
Line 136 Cryptophlebia leucotreta - should be checked, the current name of this species is Thaumatotibia leucotreta Meyrick. The species name should be written in italic.
- Response to the comment: changed in the text to the correct wording, highlighted in italics.
Line 148 it should be ‘of C. pomonella’ ?
- Response to the comment: Yes, the wording is correct. The first time the insect species name is mentioned, it must be given in full; abbreviations are allowed for subsequent mentions. Therefore, the wording "C. pomonella" is correct.
Line 251 C. pomonella - is written twice, this should be corrected
- Response to the comment: corrected in the text, removed the repeated spelling
Line 250 text in some places in Table 2 needs to be corrected
- Response to the comment: Table 2 is corrected.
Line 442 Cydia pomonella - this should be written in italics (also in other places in the Literature)
- Response to the comment: corrected the text, highlighted in italics.
Round 2
Reviewer 1 Report
Comments and Suggestions for Authors
The authors have argued the most of my major comments done to the MS. However, in the present version of the MS, they must detail how the data have been analyzed based on the Henderson-Tilton formula. Henderson-Tilton use a modification of the Abbot’s formula, and in this case it is not clear how the authors of the present MS have applied this modification.
Author Response
Thank you very much for drawing attention to this issue in our publication. We agree with your comment.
As described in the article, entomopathogenic assessment of the of C. pomonella granulosa virus strains efficacy under laboratory conditions was carried out using the Henderson-Tilton biological efficacy formula (Henderson, Tilton, 1955):
BE = (1– (B*C)/(A*D))*100
where BE is the biological efficacy expressed in reducing the pest population adjusted for control (%);
A – the number of living individuals before treatment (inds.);
B – the number of living individuals after treatment (inds.);
C – the number of living individuals in the control option in the preliminary counting (inds.);
D – the number of living individuals in the control option in the further countings (inds.).
This formula takes into account changes in the number of individuals, both in the experimental and control options. We equated the negative values obtained with zero efficacy.
It should be noted that the Henderson-Tilton formula is classical, so we did not specify it in the publication. If you consider it necessary to specify this formula, then we will insert it into the publication.